# Recent Advances in the Synthesis of Inorganic Materials Using Environmentally Friendly Media

**DOI:** 10.3390/molecules27072045

**Published:** 2022-03-22

**Authors:** Lorenzo Gontrani, Pietro Tagliatesta, Domenica Tommasa Donia, Elvira Maria Bauer, Matteo Bonomo, Marilena Carbone

**Affiliations:** 1Department of Chemical Science and Technologies, University of Rome Tor Vergata, Via della Ricerca Scientifica 1, 00133 Rome, Italy; pietro.tagliatesta@uniroma2.it; 2Department of Chemistry, University of Rome “La Sapienza”, P.le A. Moro 5, 00185 Roma, Italy; 3Department of Surgical Science, University of Rome Tor Vergata, Via Montpellier 1, 00133 Rome, Italy; donia@med.uniroma2.it; 4Italian National Research Council-Institute of Structure of Matter (CNR-ISM), Via Salaria km 29.3, 00015 Monterotondo, Italy; elvira.bauer@mlib.ism.cnr.it; 5Department of Chemistry, NIS Interdepartmental Centre and INSTM Reference Centre, University of Turin, Via Pietro Giuria 7, 10125 Turin, Italy; matteo.bonomo@unito.it

**Keywords:** Deep Eutectic Solvents, nanoparticles, inorganic synthesis, environmentally friendly media, biocompatibility, renewable feedstocks

## Abstract

Deep Eutectic Solvents have gained a lot of attention in the last few years because of their vast applicability in a large number of technological processes, the simplicity of their preparation and their high biocompatibility and harmlessness. One of the fields where DES prove to be particularly valuable is the synthesis and modification of inorganic materials—in particular, nanoparticles. In this field, the inherent structural inhomogeneity of DES results in a marked templating effect, which has led to an increasing number of studies focusing on exploiting these new reaction media to prepare nanomaterials. This review aims to provide a summary of the numerous and most recent achievements made in this area, reporting several examples of the newest mixtures obtained by mixing molecules originating from natural feedstocks, as well as linking them to the more consolidated methods that use “classical” DES, such as reline.

## 1. Introduction

According to the general definition currently used (EU 2011), a “nanoparticle” (NP) is a discrete (nano-)object where at least one of its characteristic dimensions falls in the range 1–100 nm. This category thus includes objects with a fixed number of such dimensions, such as nanowires/nanotubes (mono-dimensional, 1D) and nanodiscs/nanoplates (2D), as well as nanometric spherical/globular three-dimensional aggregates. Nanoparticles can be further classified according to their organic/inorganic nature. Dendrimers, lyposomes, and polymeric NPs belong to the former group, whereas fullerenes, quantum dots, and metal/metal oxide NPs to the latter [1,2]. Most importantly, nanomaterials are classified according to their size, shape, and properties, and it is this last feature that has led to the blossoming of nanomaterial research in the last few years. The remarkable properties nanoparticles possess (e.g., optical, magnetic, and electrical) can be exploited in a large number of technology-related fields, ranging from electronics [3,4] to biology and medicine [5,6]. The correlation between size/shape and properties in nanomaterials was thoroughly assessed in several investigations, and it was shown that their properties were dramatically dependent on the shapes, dimensions, and porosities of the synthesized nanoobjects; thus, a detailed control of the synthetic route is needed. The structural differences observed ultimately result in remarkably different performances in real-world applications. Among the factors that can be controlled, the most easily tunable is probably temperature. For instance, in the synthesis of metal oxides by the co-precipitation pathway, the temperature at which the drying/calcination of the hydroxide is carried out can vary widely. Despite its apparent “simplicity”, the effect of this variation can be quite relevant, as was found, for instance, in the synthesis of nickel (II) oxide NiO, a prototypical p-type, wide-band-gap semiconductor. NiO nanoparticles prepared at 400 °C are smaller and more porous, while larger structures are obtained at 600 °C; the band gaps associated with the latter are larger and the electrochemical performances worse, especially when these nanoparticles are introduced into screen-printed electrodes (SPEs) and employed as electrochemical sensors for selective assays or in Faradic pseudo-capacitors. Other factors that have been found to influence the intercalation of precursors and the sintering of particles, the latter leading to particles of larger sizes, are the reagents used to prepare the precursor hydroxide Ni(OH)_2_: the use of various combinations of nickel salts and inorganic/organic bases as precipitating agents, and the adoption of surfactant-free hydrothermal synthesis or microwave irradiation (which can lead to different results) [7,8,9,10,11,12].

In a large number of studies aiming to control the shape and dimensions of particles, the use of templating agents has been investigated in more detail. Various types of systems have been employed for this purpose, such as, for instance, boric, citric, and ascorbic acids [13]; cholesteric liquid crystals [14]; and different types of surfactants [15]. Here, the use of dendrimers such as poly(amidoamine) (PAMAM) is noteworthy, as it allowed the production of very small NPs with dimensions as low as 1.4 nm [16]. In this contribution, we focus our attention on a new class of environmentally friendly, inherently inhomogeneous, and highly structured family of liquids, Deep Eutectic Solvents, which can naturally exert a templating effect. The most recent developments in the field will be reviewed and additional references will be provided, thus enlarging the literature coverage of the reviews already published on this subject [17,18,19,20].

## 2. Brief Description of Deep Eutectic Solvents

Since the “official” birthday of this family of liquids, which can be traced back to the pivotal paper by Abbott in 2003 [21], a fairly large amount of research has been dedicated to them, especially in the last decade. The key feature of these solvents is their very easy and economic preparation, which involves the simple mixing of at least one hydrogen bond donor (HBD) with at least one hydrogen bond acceptor (HBA). In most cases, this takes place in solid phase and in definite ratios, providing mixtures with remarkably lower melting points than those of their individual components. The sometimes very large decrease seen in the melting temperature, which in the prototypical choline chloride:urea 1:2 system (“Reline”) [22] can be more than 100 °C with respect to the ideal value obtainable from thermodynamics arguments [23,24], has led to the term “deep” being used to distinguish DESs from other non-ideal physical systems where similar phenomena are observed—e.g., water–DMSO mixtures. Deep Eutectic Solvents were divided into four main classes in the original classification by Abbott, three of which contain a halide anion and a quaternary ammonium cation as HBA (in most cases, choline chloride), and differ in their type of HBD: metal chloride in class I, hydrated metal chloride in class II, organic molecules (such as urea, glycerol, and carboxylic acids) in class III (which is the most populated), and metal halides and urea (or acetamide/ethylene glycol) in class IV. Recently, a new class was introduced (“type V” [25]), containing only hydrogen-bond donors and acceptors and no ions. Other non-negligible valuable traits of Deep Eutectic Solvents are their high sustainability, especially in the subfamily NaDES (Natural DES), where the precursors are benign compounds generally obtained from natural feedstocks, such as choline chloride and carboxylic acids; these therefore have a low toxicity and high biocompatibility and biodegradability [26,27,28]. Additionally, DESs are characterized by their high conductivities, viscosities, and surface tensions; have a low volatility and flammability; and have highly tunable physiochemical properties, considering the large number of possible combinations of precursor salts [29]. These last features are shared with ionic liquids (ILs), which can be considered as the forefathers of DESs. However, ILs are composed (almost entirely) of ions [30,31], whereas DESs, apart from the type V ones, are mixtures that contain polar molecules and ions. A sketch of the desirable technological properties of DESs is provided in Figure 1.

## 3. DES in Nanoparticle Synthesis from Their Birth to 2020

The investigation of the applicability of ILs and DESs in the field of nanoparticle synthesis started with a focus on the former family, but the amount of research on DES has been growing lately. In addition, some of the features characteristic of ILs can be extended seamlessly to DESs. The most striking property of DESs in the field of nanoparticle synthesis is the high solubility of metal ion salts and complexes in several cases. As already shown by Abbott et al. [32], the solubility of some bulk metal oxides, which are almost insoluble in pure water, is greatly increased in some DESs, reaching values of up to more than 10,000 parts per million in ChCl:malonic acid 1:1 at 50 °C for Cu_2_O, CuO, and ZnO, and up to 90,000 ppm for ZnO in ChCl:urea 1:2 at 70 °C. In contrast, other oxides, such as iron oxides, are barely soluble in ChCl:urea 1:2. This feature can be used, for instance, to design suitable synthesis protocols based on fractional precipitation. Another key factor is the possibility of these substances acting in a dual capacity as both solvents and templates for nanostructure formation (“target solvent”) [33,34,35,36,37], owing to their structural heterogeneity [38,39,40,41]. In several cases, DESs can behave as reactants as well [42,43,44].

The heterogeneity and microstructure of DESs have attracted great interest. The study by Chen et al. [45] pointed out that in “reline” (ChCl:urea 1:2), “ethaline” (ChCl:ethylene glycol 1:2), and “glyceline” (ChCl:glycerol 1:2 (the classification of this mixture as a DES or as a salt-in-solvent system has recently been debated, and the data are in favor of a 1:3 composition defining a proper DES [46])), the charged and neutral portions of the mixtures tend to separate when the liquid comes into contact with graphite electrodes, with choline and chloride ions being attracted into the Stern layer, while the molecular components are excluded at all the potentials scanned. Similar experiments carried out using Pt(111) electrodes and the same three DESs were conducted at various potentials and at increasing water contents [47]. This study showed that the addition of water in amounts up to ≈40 wt% did not lead to the decomposition of the interfacial nanostructure, as observed in ILs even at very small water contents. Yet, the cyclic voltammetry analysis reported in the same study indicates that, at such high water concentrations, the nanostructure appears comparable to that of salt solutions. The preservation of the microscopical structure upon hydration was also demonstrated in a neutron diffraction study conducted by Hammond et al. [48], who pointed out that the microscopic features of pure reline were maintained in a water to DES molar ratio of up to 10:1 (around 42 wt% H_2_O); beyond that limit, the system can be better described as a three-component HBA:HBD:water mixture. Marked heterogeneity was evidenced as well in a molecular dynamics study conducted by Alizadeh et al. [34], who reported the fate of ethaline’s polar and nonpolar molecular groups under electrostatic potential. The calculations suggest that the DES reorganizes in order to maximize the interactions between domains of the same polarity, thus inducing strong heterogeneity in the system.

The oldest syntheses reported mainly concerned the preparation of noble metal nanoparticles with a controlled shape; in most cases, these syntheses employed Abbot’s DES reline [21]: catalysts containing star-like gold NPs were synthesized through the reduction of HAuCl_4_ by L-ascorbic acid in DES without surfactants or seeds at room temperature ([49], see Figure 2). The shape could be further changed (e.g., to a snowflake or nanothorn) by adjusting the water content. Other noble metal NPS were prepared by electrosynthesis methods: in this regard, DESs can be very valuable, since they possess very wide electrochemical windows just slightly bit smaller than ILs which can be effectively coupled to the large solubility of metal oxides to set up electrochemical cells for the deposition of metal nanoparticles, achieving very efficient control over the nucleation, deposition rate, and size of the crystals obtained [50]. A further very important aspect of noble metal deposition is that DES and ILs can substitute highly toxic cyanide-based electrolytes [51]. Among these examples, the preparation of concave tetrahexahedral Pt nanocrystals by electrodeposition using reline without employing surfactants, seeds, or other chemicals but with a high control of the shape and a high surface energy was reported [52]. Two-dimensional superstructures of aggregated Pd nanoparticles were electrodeposited from choline chloride:urea 1:2 onto glassy carbon foil, with the absorbed species forming an anionic layer that was observed with Ultra Small X-Ray Scattering (USAXS) [53]. Platinum icosahedral nanocrystals with high-index facets and a higher electrocatalytic activity and stability were electro-synthesized in reline [54]. Further details on the role of Deep Eutectic Solvents in the synthesis of plasmonic (Au, Ag, Pt) nanoparticles can be found in the recent review of Des et al. [55], who also describe biocompatible capping strategies that make use of polysaccharides (carrageenan), resulting in highly monodisperse nanoparticles. Other non-electrochemical redox reactions were carried out using environmentally friendly routes based on reline as a solvent: CuCl nanocrystal powder, which is a very useful catalyst for organic synthesis, was obtained either through the synproportion of CuCl_2_ and Cu [56] or by the reduction of CuCl_2_ by ascorbic acid [57]. Additionally, spherical, magnetic nanoparticles of ferrous ferrite (Fe_3_O_4_) were prepared at 80 °C and successfully tested for the absorption of Cu^2+^ ions, proving to be superior to NPs prepared in pure water [58]. Further examples of systems prepared in reline with co-precipitation are PbS nano/micro superstructures made from lead (IV) and thioacetamide [59], mesoporous NiO [60], and some examples of iono-thermal reactions at high temperatures/pressures (nanoflower-like α-Ni(OH)_2_ and NiO [61], Ni_2_P supported on amorphous/mesoporous Ni_3_(PO_4_)_2_-Ni_2_P_2_O_7_ [62], nanosized SnO crystals [63], Fe_2_O_3_ nanospindles as high-capacity anode materials [64], mesoporous Co_3_O_4_ sheets or nanoparticles [65], and MnCO_3_/MnOx mesocrystals [66]). An interesting example of an “antisolvent approach” is the synthesis of ZnO nanocrystals doped with Cu(II) ions, which involves adding a controlled amount of water to solutions of bulk ZnO in DES [67]. Few studies have employed DESs other than reline; some noteworthy examples utilize ethylene glycol as the HBD partner of ChCl, the latter being used to obtain nanocrystalline SnO_2_ or SnO_2_/graphene nanocomposites [68], nanoporous Ag films [69], and Ni-P alloy nanoparticles [70] to deposit Ni deposits [71], Ni-Ti nanocomposite coatings [72], or Ag films [73]. In an interesting study by Mota-Morales [74], the dispersant properties of DESs during the formation of nanoparticles and other nanocomposites were exploited for the preparation of a microporous carbon nanotube-polyacrylic acid composite using a ChCl:acrylic acid mixture. The latter DES was also successfully employed as a functional monomer to create DES-levofloxacin-imprinted Pd nanoparticles for the selective removal of pollutants from wastewater [75].

More recently, the increasing need for even greener and more biocompatible alternatives has fostered the usage of new types of environmentally benign and low-cost mixtures, mostly belonging to the NADES family (DESs of natural origin), where the precursors are obtained from renewable feedstocks. For instance, Zainal-Abidin et al. recently reported that graphene is significantly less cytotoxic when it is functionalized with ChCl:glucose/fructose/sucrose) 2:1 or, better yet, ChCl:malonic acid 1:1, with respect to pristine or oxidized graphene owing to surface modifications [76]. NADES mixtures containing glucose, fructose, and sucrose as HBD, choline chloride, and water at different molar ratios, were employed in the synthesis of MoS_2_ nanosheets by Mohammadpour et al. [77]. The material prepared was stable in aqueous environments, could perform as a catalyst in hydrogen evolution reactions (HERs), and could be obtained in a higher yield compared to other exfoliating agents (average of 44% vs. 20%). New mixtures created by changing the HBA or HBD started to be explored in the last part of the 2010s: an interesting example of those belonging to the second group is the synthesis of calcite nanoparticles by the reaction of CO_2_ with “calcoline”, a DES composed of choline chloride and calcium chloride [78]. This study demonstrates that DESs can be successfully exploited in “carbon-reduction” protocols, leading to value-added products. A modification of the acceptor moiety (HBA) was used by Adhikari et al., who reported the use of halide-free DES, where the Cl-anion of reline is replaced by nitrate, for the microwave-assisted reduction of silver salts to organic-soluble oleylamine-capped Ag nanoparticles [79]. Following on from this study, Adhikari et al. later fine-tuned a silver-based DES (1:4 silver triflate:acetamide, “argentous DES”) that allowed them to obtain large amounts of monodispersed colloidal silver nanocrystals of high quality despite the high metal concentration, owing to the “size focusing” effect of the DES that suppressed uncontrolled nanocrystal growth [80]. More recently, these researchers exported their methodology to flow-reactor synthesis employing dimethylammonium nitrate-polyol DES media; they were able to obtain a 1000- to 4000-fold increase in throughput compared to conventional synthesis [81]. Other noteworthy examples of HBA modifications include mixtures of non-quaternary cations, such as dimethylamine, ethylamine, and trimethylamine hydrochlorides with urea, which have been successfully used to synthesize a series of silver and selenido-stannate ([NH_2_(CH_3_)_2_]_2_Sn_3_Se_7_·0.5NH(CH_3_)_2_, [NH_4_]_2_Sn_4_Se_9_, [NH_3_C_2_H_5_]_2_Sn_3_Se_7_, and [NH_4_]_3_AgSn_3_Se_8_) crystals through the solvothermal pathway. The ammonium cations originating from the original HBA as well as those resulting from the decomposition of urea have been proven to be able to act as templating agents, leading to the establishment of three-dimensional supermolecular inorganic frameworks that show the peculiar feature of thermochromism in some cases [82]. The use of ammonium cations as hydrogen bonds, though very common and convenient, is not the only option. In fact, other inorganic salts have recently been used. For example, ionic compounds belonging to the alkali halide family have been proven to form polyol-based DES mixtures (such as CsF/KF:glycerol) that have shown very high selectivities when used as reaction media for copper-catalyzed homocoupling organic reactions [83].

Returning to DESs containing molecules of natural origin, a mixture of caffeic acid with ChCl and ethyleneglycol was used to prepare moleculary imprinted hexagonal boron nitride NPs, which were successfully employed in the solid-phase extraction of flavonoids [84], while tartaric acid was used as a DES component in a recent electrodeposition preparation of a Ti/SnO_2_–Sb electrode with a high electrochemical activity [85].

In summary, the main advantages of DES that were readily highlighted shortly after their introduction as synthesis media and that were largely demonstrated during this initial time period lie in their friendliness towards ecosystems, their highly tunable physiochemical properties, and their cheap means of preparation and handling. An additional very important and profitable feature is their intrinsic microheterogeneity, with it being possible to confer specific morphologies to the obtained nanoparticles. A possible drawback of the use of DESs is their moderate viscosity, which depends on the nature of their components, with the viscosity being lower for hydrophobic DESs and larger for sugar-based NADESs. Indeed, it has been shown that the decrease in mass diffusivity caused by viscosity affects nanoparticle growth and generally leads to NPs of a larger size [86].

## 4. Research on DES Blooms: From 2020 up to Now

The number of investigations on a multifaceted topic such as DES have increased markedly in the last few years. The simple query “Deep Eutectic Solvents” in the title, abstract, and keywords on Scopus database yielded 6474 results on 17 February 2022; a more refined inquiry (“deep eutectic solvents” AND “nanoparticles”) on the same day led to 343 results, 22 of which were published in the first fifty days of 2022. The full set of trends discovered is shown below (Figure 3). During the last two years, research in this field has continued in the direction of considering new eutectic mixtures that are often cheap and biocompatible. The published studies include “wet chemistry” synthesis protocols (i.e., synthetic routes not assisted by electrochemical methods or other instrumentations, such as CVD; see Table 1 for a schematic report) as well as electrochemical deposition methods, which have already been proven to be very efficient in producing several types of nanoparticles. Among all the literature, although reline still plays an important role (since it features in approximately one out of four studies (80/343) on both traditional liquid-phase syntheses and electrodeposition), the interest in new methods has grown significantly and NADESs are becoming more and more important every day. Among the newest and most interesting investigations using “traditional” ChCl:urea, the latter DES was used: in the field of metal oxides synthesis, to prepare nontoxic photoluminescent SnO nanoparticles for cell imaging [87], in the solvothermal synthesis of iron oxide from iron monitored in operando by SANS and EXAFS [88], in the synthesis of ceria nanoparticles in a continuous microreactor [89], to conjugate Fe_3_O_4_ nanoparticles on graphene oxide [90], to obtain NixCo_2_-x(OH)_3_Cl nanoparticles with an optimal Ni/Co ratio suitable for use as cathodic materials [91], as a solvent and nitrogen source for preparing fluorescent carbon dots [92], and for the synthesis of titanomagnetite NPs with enzyme-like activity [93]. Finally, reline was also used very recently as a template to prepare polyacrylates/nanohydroxyapatite scaffolds and polycrystalline sub-nanometrical chalcogen nanoparticles (SeTe) with high activity towards tumor cell lines [94].

Regarding electrodeposition studies conducted in reline, several examples have been published in the last two years: for instance, water-soluble magnetic iron oxide nanoparticles were prepared at the anode during the electrolysis of reline using iron electrodes [95]; ruthenium nanoparticles were electrodeposited on stainless steel from a dichloride solution [96]; core–shell Pd-hydroxide NPs were obtained through the electrochemical oxidation of formic acid [97]; Zn/ZnO nanopowders were electrodeposited over Pt electrodes electrolyzing ZnCl_2_ solutions [98]; copper nanoparticles were obtained from CuCl_2_·H_2_O [99]. Some of the other “traditional” mixtures prepared in the first phase of DES development together with reline—namely, ethaline (choline chloride-ethylene glycol) and glyceline (choline chloride-glycerol)—have found a considerable number of applications recently. Ethaline was used as a medium for the electrodeposition of Ni nanoparticles. It was, suitable for the catalysis of methanol oxidation [100]; for the synthesis of NiS_2_ nanospheres [101]; and, very interestingly, for the modification of Fe_2_O_3_ NPs to produce very specific electrochemical sensors—for H_2_O_2_, acetylcholine and the antibiotic dapsone [102]. Furthermore, ChCl:EG was used as a carrier and disperser for SiO_2_@Fe_3_O_4_ (silica-coated magnetite) ferrofluids selective for Meloxicam [103], to electrodeposit Ni-Al nanocomposite coatings from NiCl_2_ and aluminum dispersions [104], to synthesize platinum hollow-opened structures with enhanced performance in the electro-chemical oxidation of methanol [105], and as a solvent for assembling Ag nanoparticles onto copper substrates [106]. Glyceline was employed in the solvothermal preparation of functionalized metal-organic framework (MOF) nanoparticles containing zirconium [107], to immobilize Pd nanoparticles [108], and to deposit ZnO in situ on graphene sheets [109]. However, the real novelty of the last period consisted, as stated above, in the very large increase in the use of many new and alternative mixtures, often including biomolecules of natural origin. For instance, the excellent capabilities of glycerol as hydrogen bond donor in DES were further enhanced by the addition of malic acid and D-fructose in a 1:1:1: mole ratio (MaFruGly). MaFruGly was used to disperse Al_2_O_3_ into a nanofluid that is capable of extracting polyphenols and other bioactive compounds from olive oil pomaces and leaves [110]. Other examples of such biocompatible mixtures include choline chloride:glucose, which was used to prepare sodium hyaluronate/dopamine/AgNPs hydrogels [111]; ChCl:xylitol, which was employed to modify magnetic titania NPs with Fe_3_O_4_@TiO_2_@DES [112]; and choline chloride:gluconic acid DES, which was used to prepare a cobalt-DES precursor that was finally pyrolyzed into Co nanoparticles supported on nitrogen-doped porous carbon (Co@NPC). The latter acts as a bifunctional catalyst for water splitting (H_2_ production) and glucose oxidation (GOR) in an electro-chemical cell [113]. Along the same line, carbon-doped copper oxide catalysts with a high selectivity in CO_2_ electrochemical reduction were produced by the calcination of sugar-urea DES (glucose:urea and galactose:urea) containing copper salts [114]. Mixtures containing carboxylic acids have been described in several examples: ChCl + oxalic acid and malonic acid, together with urea, fructose, and ethylene glycol, were used to synthesize MgFe_2_O_4_ nanoparticles with a high sensitivity and selectivity towards nitrofenantoine and 4-nitrophenol [115]. Pd nanoparticles confined in nanocellulose with a high catalytic chemoselectivity and activity were prepared from choline chloride and oxalic acid dihydrate 1:1 [116] (this mixture was also employed as a medium to synthesize porous Fe_3_O_4_ nanosheets with high electrocatalytic performances starting from commercial powders) [117]. An example of a mixture of choline and inorganic salts is choline:Na_2_SO_3_ 2:1, a “reductant” solvent where sulfur-functionalized graphene oxide NPs for Li-S batteries were synthesized using a chemical reduction/co-precipitation method [118].

A relevant portion of the recent studies on this topic has focused on magnetic nanoparticles containing Fe_3_O_4_ (or Fe_2_O_3_) and organic moieties or enzymes, which could be assembled in a few examples of DESs: acrylic acid-Fe_3_O_4_ composites were obtained from acrylic acid:menthol-type V DES used in the detection of pesticides [119], whereas macroporous polyacrylamide γ-maghemite composites were prepared in acetic acid:menthol [120]. Nano-Fe_3_O_4_ was prepared in NADES betaine-urea, coated with silicon, and successfully employed to immobilize β-glucosidase [121]. magnetite nanocubes with anticancer and photo-Fenton efficacy were synthesized in ChCl:citric acid by Sakthi Sri et al. [122]; poly glycerol@Fe_3_O_4_ nanoparticles (as HBD) were treated with choline chloride, giving a branched poly (DES)@ Fe_3_O_4_ fluid that was used as draw solute in forward osmosis [123], whereas DES coupled to Fe_3_O_4_-MUiO-66-NH_2_ (a metal organic framework composed of zirconia clusters cross-linked by terephtalic acid) obtained from mixtures of quaternary ammonium salts different from choline, such as benzyltributylammonium chloride (BTBAC), benzyltributylammonium bromide (BTBAB), tetrabutylammonium chloride (TBAC) as HBA, and lactic or glycolic acid such as HBD was employed to adsorb pharmaceuticals and personal care products (PPCPs). The magnetic nanoparticles could be easily separated and recovered from the adsorbed species using a magnet [124]. Other examples of quaternary ammonium salts include TBAB (tetrabutylammonium bromide), which was combined with imidazole in a DES. The latter was employed as a solvent to immobilize Pd nanoparticles onto a covalent organic framework, finally resulting in a heterogeneous catalyst used for the phosphorylation of aryl bromides [125]. Furthermore, CTAB (cetyltrimethylammonium bromide), which combines with acetic acid in a 1:1 mass ratio to create a liquid mixture, was used for the formation of ceria nanoparticles with remarkable photocatalytic activity by Iqbal and coworkers [126,127]. Other sugar-based DESs similar to those cited above—namely, dl-menthol:oleyl alcohol 1:1.2—were employed to prepare an NDDES (nano dispersed DES) mixture containing boron nitride nanoparticles, with excellent properties as a heat-transfer nanofluid [128]. A comprehensive survey on the use of DES as a base fluid for heat-transfer nanofluids can be found in [29].

**Table 1 molecules-27-02045-t001:** Recent examples of “wet syntheses” in DES media.

Solvent	Reagents/Path	Product	References
ChCl:urea 1:2	Preparation of separate solutions of HAuCl_4_⋅4 H_2_O (0.015 g) and L-ascorbic acid (LA, 0.05 g) in DES. Addition of LA solution to HAuCl_4_ at 30 °C under magnetic stirring until the color changes from yellow to dark purple	**Au** nanoparticles Star, snowflake, or nano-thorn-shaped depending on water content	[49] (Figure 2)
	Direct electrodeposition on GC substrate in 19.3 mM H_2_PtCl_6_/DESs solution at 80 °C	Tetrahexahedral (THH) concave **Pt** NCs	[52]
		Triambic Icosahedral **Pt** NCs	[54]
	Mixing CuCl_2_·2H_2_O (5.0013 g, 0.0293 mol) and Cu powder (1.6935 g, 0.0265 mol) with DES, gentle stirring at 20 °C for 5 h, rinsing with diluted HCl	**CuCl** nanocrystal powder	[56]
	Addition of 2.2232 g of CuCl_2_ 2H_2_O and 1.3754 g of ascorbic acid to 14 mL DES in the presence of PVP, mild stirring at 25 °C for 1 h, rinsing with 50 mL HCl (0.1 M)	Spherical **CuCl** nanoparticles	[57]
	Addition of 2.164 g (8 mmol) FeCl_3_·6H_2_O and 1.194 g (6 mmol) FeCl_2_·4H_2_O to 15.585 g DES, stirring at ca. 600 rpm and 80 °C for 20 min, subsequent addition of 2.613 g (46.7 mmol) KOH, and stirring for another 1.5 h at 80 °C. Alternatively, see Figure 5 and last paragraph	Spherical, magnetic **Fe_3_O_4_** nanoparticles	[58]; Figure 5 and last paragraph
	Solvent: 35.70 g of DES (30 mL at 37 °C) and 6 mL of water. Dissolution of thioacetamide (TH, 12 mmol, 0.9134 g) mL into 12 mL of solvent and lead (IV) acetate (LAC, 12 mmol, 5.3206 g) into the remaining liquid, stirring of both solutions at 80 °C. Injection of TH into LAC changes the solution from pale yellow to opaque dark brown. Rinsing with water, followed by dialysis, centrifugation, and drying in furnace at T ≤ 80 °C	Hyperbranched **PbS** Nano/microcrystals	[59]
	Heating of NiCl_2_·6H_2_O in DES (0.1 M solution) at 150 °C for 40 min; then addition of 10 mL of water and further stirring for 20 min, cooling in ice bath; drying of precursor overnight at 90 °C and further annealing in air (300 °C) for 4 h	Mesoporous **NiO**	[60]
	NiCl_2_·6H_2_O in DES ionothermal reactions at different temperatures and conditions	**Ni(NH_3_)_6_Cl_2_**, **NiCl_2_** and nanoflower-like α-**Ni(OH)_2_** and **NiO**	[61]
	Dissolution of 5.94 g of Ni(H_2_PO_2_)_2_·6H_2_O (0.02 mol) and 1.66 g of NH_4_H_2_PO_2_ (0.02 mol) in 27.92 g (0.2 mol) of choline chloride and 24.02 g (0.4 mol) of urea, stirring at 323 K under N_2_ for 30 min, reduction of product with H_2_ at 673 K for 3 h	Ni_2_P supported on amorphous/mesoporous **Ni_3_(PO_4_)_2_-Ni_2_P_2_O_7_**	[62]
	Emulsion of 2.25 g SnCl_2_·2H_2_O in 100 mL DES. Variable reaction times (1 to 60 min)	Nano-sized **SnO** particles (20–30 nm)	[63]
	Heating of 40 mL of 0.1 M FeCl_3_·6H_2_O/DES solution at 200 °C, after 10 min. addition of 40 mL of water and further reaction for 10 min. Washing of precipitate with ethanol and dried at 80 °C overnight	**Fe_2_O_3_** nanospindles	[64]
	Dissolving CoCl_2_·6H_2_O into ChCl to obtain a 0.1 M CoCl_2_:ChCl solution, addition of 100 mL of water after heating for 40 min at different temperatures. Subsequent ice bath cooling, rinsing of product with water and methanol, and drying at 70 °C under vacuum	Mesoporous **Co_3_O_4_** sheets or nanoparticles	[65]
	DES solution of bulk ZnO. Precipitation with water (anti-solvent approach)	**ZnO** nanocrystals doped with Cu(II) ions	[67]
	Dissolution of SnCl_2_·2H_2_O in DES, stirring in pre-heated water bath (50, 80, 98 °C), precipitation with ethanol, and drying at 230 °C	**SnO_2_** nanoparticles	[68]
	Mixing of y mmol NiCl_2_ 6H_2_O and 20-y mmol CoCl_2_ 6H_2_O (*y* = 0, 2.5, 5, 10) in 10 mL DES. Addition of 1 mmol SDS and 20 mL water, heating for 12 h at 100 °C, washing of precipitate with water and ethanol	**NixCo_2_-x(OH)_3_Cl**	[91]
	Stirring of 1.668 g (0.006 mol) FeSO_4_·7H_2_O and 0.584 g (0.010 mol) of KOH 0.408 g in DES for 30 min, addition of 0.408 g (0.0012 mol) tetrabutyl titanate (TBOT) and 0.420 g (0.008 mol) of KOH, stirring first at 80 °C (30 min) and then at 110 °C (4 h), washing of precipitate with water and ethanol	**Fe_2.5_Ti_0.5_O_4_-DES** nanoparticles	[93]
ChCl:urea 1:2 ChCl:urea:water 1:2:10	Hydrothermal treatment of Fe(NO_3_)_3_·9H_2_O/DES mixtures (dry and hydrated DES) for 3–8 h at 90 °C before particles are dried at 60 °C from ethanol after dialysis	**FeO**	[88]
	Dissolution of Ce(NO_3_)_3_·6 H_2_O in DES and stirring at 250 rpm for 40 min, reaction in pressurized continuous microreactor at 100–160 °C, washing of the solid product with water and ethanol and drying at 80 °C	**CeO_2_**	[89]
(CH_3_)NH_2_ HCl:urea 1:1.5	Mixture of Sn (0.119 g, 1.0 mmol), Se (0.211 g, 2.67 mmol), dimethylamine hydrochloride (0.58 g, 7.1 mmol), urea (0.64 g, 10.67 mmol), and 0.3 mL of N_2_H_4_·H_2_O (98%) (∼6.17 mmol), hydrothermal synthesis at 160 °C (3 h), rinsing with water	Silver and selenido-stannates **[NH_4_]_3_AgSn_3_Se_8_ [NH_4_]_2_Sn_4_Se_9_** **[NH_3_C_2_H_5_]_2_Sn_3_Se_7_**	[82]
ChCl:oxalic acid 1:1	Dissolution of 30 mg of commercial Fe_3_O_4_ in 1 mL ChCl/OA DES at 50 °C by ultrasonic treatment, microwave heating for 10 s at 100 W, further thermal treatment at 300 °C for 2 h	**Fe_3_O_4_** nanosheets	[117]
	Addition of MgO and α-Fe_2_O_3_ to DES molar ratio 1:1 (0.5 wt% melt in the overall amount of metal oxides), stirring for 1 h, then calcination of melts at 500 °C for 1 h (5 °C min^–1^ heating rate)	**MgFe_2_O_4_** nanoparticles	[115]
ChCl:acrylic acid	Stirring of ChCl and MAA in the molar ratio 1:2 at 80 °C; mixing with a porogen (MeOH), initiator (AIBN), crosslinking agent (EGDMA), and template (levofloxacin); heating at 60 °C for 12 h; removal of template by Soxhlet extraction with methanol	Levofloxacin-imprinted **Pd** nanoparticles	[75]
ChCl:oxalic acid:water 1:1:1	Mixing of cellulose pulp (0.5 g) with DES (10 g) and water (10 g), heating at 110 °C for 2 h in a Teflon-lined reactor to obtain carboxylic cellulose (CNF). Addition of 10 mL of PdCl_2_ (17.7 mg) in HCl and aqueous NaBH_4_ (10 mg, 1 mL) to a diluted CNF suspension (20 mL, 0.4 wt%), reaction at 4 °C for 4 h, separation of Pd NPs by dialysis	**Pd** nanoparticles confined in nanocellulose	[116]
ChCl:ethylene glycol 1:2	Mixing of NiSO_4_·6H_2_O (0.1 M), Na_2_S_2_O_3_·5H_2_O (0.1 M), EDTA (0.06 M), and DES in a beaker at different temperatures (80 °C, 100 °C, 110 °C, 120 °C, 160 °C); stirring of the mixtures for 3 h; washing of the solids with water and ethanol; and drying at 60 °C	**NiS_2_** nanospheres	[101]
	Dissolution of 4.0 mg of Pt(acac)_2_, 40 mg PVP, and 25 mg of SDS in 8 mL DES; heating in oil bath at 130 °C for 2 h; washing of the black precipitate with ethanol	**Pt** hollow-opened structures	[105]
ChCl:glycerol 1:2	Hydrothermal heating of ZrCl_4_, BDC (1,4 benzene dicarboxylate), H_2_O, and DES at a molar ratio of 1:1:1:500 at 120 °C for 48 h; washing of the solid with water	Nanoparticles containing **ZrCl_4_**	[107]
	Mixing 2.19 g of Zn(CH_3_COO)_2_·2H_2_O and 0.2 g of graphene in 50 mL DES, precipitation with 0.8 g NaOH	**ZnO** in situ on graphene sheets	[109]
ChCl:CaCl_2_ 1:2	CO_2_ capture from air of CaCl_2_·6H_2_O and choline chloride DES at 50 °C under stirring at 400 rpm, formation of CaCO_3_ sediment after 6 h, washing of the sediment with water, drying at 60 °C for 12 h, reuse of the filtrate for further CO_2_ capture	**CaCO_3_** NPs	[78]
ChCl:glucose/fructose/sucrose/maltose/raffinose	Liquid-phase exfoliation of MoS_2_ in glucose, fructose, sucrose, raffinose, maltose, choline chloride, and water DES at various ratios (5 mg MoS_2_ per mL of DES); separation of exfoliated material in ethanol/water	**MoS_2_** nanosheets	[77]
CHCl:glucose	DASH: Dopamine hydrochloride (DA), N-Hydroxysuccinimide (NHS), 1-ethyl-3-(3-(dimethylamino)propyl) carbodiimide (EDC), sodium hyaluronate (SH) in 2-(N-morpholino) ethanesulfonic acid-buffered solution (MES buffer). Addition of AgNO_3_ to DES-DASH 4:175 mixture	**Na hyaluronate/dopamine/Ag NPs** hydrogels	[111]
ChCl:xylitol 1:1	Mixing 0.2 g Fe_3_O_4_@TiO_2_ nanoparticles and 3.0 mL [ChCl][Xyl] by ultrasonication for 2 h, separation by external magnet, rinsing with water	**Fe_3_O_4_@TiO_2_@DES**	[112]
ChCl:gluconic acid	Mixing 2 g choline chloride, 4 g urea, and 0.4 g Co(NO_3_)_2_·6H_2_O in 5.62 mL of 50% gluconic acid solution; calcination in N_2_ at 700–900 °C after freeze-drying	**Co@NPC**	[113]
CHCl:citric acid 2:1	Addition of 3.9813 g FeCl_2_ 4H_2_O (20 mM) and 8.1091 g of FeCl_3_.6H_2_O (30 mM) at the molar ratio 1:1.5 to DES, stirring at 80 °C (600 rpm) for 20 min, addition of 40 g (712.94 mM) KOH, stirring for another 1 h, washing with ethanol and water	**Fe_3_O_4_** nanocubes	[122]
Betaine-urea 1:2	DES: betaine (2.343 g) and urea (2.4 g), heating for 15 min at 125 °C, addition of 1.5 mL water, dissolution of 0.111 g FeSO_4_·7H_2_O (0.4 mmol) and 0.216 g FeCl_3_·6H_2_O (0.8 mmol) in DES at RT under stirring (10 min), precipitation by the addition of 0.2 g of KOH (3.5 mmol), separation with external magnet, and washing with water	Nano-**Fe_3_O_4_** Nano-**Fe_3_O_4_**@SiO_2_–NH_2_	[121]
BTAB/BTBAC/TBAC:lactic acid	[BTBAC][Lac]-DES: Mixing 3.12 g BTBAC and 1.80 g Lac at a molar ratio 1:2 under heating at 80 °C in oil bath for 1 h. Addition of 2.0 mL DES to a phosphate buffer (20 mM, pH = 7.0) containing 0.24 g of NHS and 0.16 g of EDC·HCl to activate the carboxyl group of DES; subsequently, the addition of 0.20 g MUiO-66-NH_2_, stirring for 12 h, washing of particles with water, and freeze-drying	**Fe_3_O_4_**-MUiO-66-NH_2_	[124]
TBAB:imidazole	Condensation of TFPT (main building block) and hydrazine (comonomer) in BuN_4_Im/Br at 90 °C for 12 h, subsequent impregnation with Pd(oAC)_2_ under reflux	**Pd@MOF**	[125]
CTAB:acetic acid 1:1	Mixing cetyltrimethylammonium and acetic acid at 70 °C for 3 h. Addition of 1 g ammonium cerium (IV) nitrate to 0.5 g of DES and hydrothermal treatment of the solution at 130 °C for 7 h. For N-doping, a urea solution (10 g/30 mL of water) is added, followed by the separation of particles by centrifugation and washing with ethanol and acetone	Plain and N-doped **CeO_2_**	[126,127]
dl-menthol:oleyl alcohol 1:1.2	Mixing 1 mol D,L-menthol and oleyl alcohol at 343.15 K under stirring for 12–24 h. Addition of h-BN nanoparticles at different weight percentages, shaking, and sonication for 2 h	**BN** nanoparticle nanofluid	[128]
Acetic acid:menthol 1:2 pyruvic acid:menthol 1:1 lactic acid:menthol 1:2 lauric acid:menthol 2:1	Mixing of D,L-menthol with PA, AA, LacA, or LauA at 50 °C for 15 min before drying under vacuum (10^−1^ Pa). Preparation of high-internal-phase emulsions (HIPEs) by dropping DES into a continuous phase of AAm:BAAm (acrylamide:N,N′-Methylenebis(acrylmide), polymerization with potassium persulfate (KPS), and coating with γ-Fe_2_O_3_	Polyacrylamide γ-maghemite composites	[120]
Acrylic acid:mentho 1:2	Mixing AA and menthol at 70 °C in a water bath for 5 min, polymerization of DES via a thermal frontal method usingFe_3_O_4_ NPs-AA as a cross-linker and thermal initiator into a magnetic poly (AA-menthol DES) hydrogel	**Acrylic acid**:**Fe_3_O_4_** composites	[119]
Choline:Na_2_SO_3_ 2:1	Heating of 7.0 g of choline chloride and ∼4.0 g NaS_2_O_3_ in 2:1 molar ratio at 40 °C for 3 h. Addition of 16 mL DES to a GO solution in the presence of hydrazine as a reducing agent, co-precipitation of reduced GO and sulfur	Sulfur-functionalized **graphene oxide** NPs	[118]
Dimethylammonium nitrate:triethylene/ethylene glycol, or glycerol 1:1	HBA: Addition of 119.4 mL of 5.0 N HNO_3_ solution (0.597 mol) to aqueous dimethylamine (40 wt% in H_2_O, 0.597 mol). HBD: triethylene glycol, ethylene glycol, or glycerol. Addition of HAuCl_4_ or AgNO_3_ and oleylamine (OAm) (reducing agent) in each DES under stirring (150 rpm); heating at different temperatures (12 h at 60 °C for Ag, 19 min. 140–170 °C for Au)	**Ag** or **Au** colloidal nanocrystals	[81]

## 5. Lewis Acid DES (LADES)

A smaller number of studies were dedicated to DES without quaternary ammonium salts, containing metal salts (often hydrated) such as HBA and hydrogen bond donors such as urea or acetamide. This family of DES is known as LADES (Lewis Acid DES)) [129], compared to the more common “BADES” (with Brønsted acids, such as choline chloride + oxalic acid). According to the Abbot classification, LADES belong to Type I, II, or IV DES (such as, for instance, ChCl:2ZnCl_2_, ChCl:2CrCl_3_·6H_2_O, ZnCl_2_:urea 1:3.5 (or:4) melts), as do several amide-nitrate eutectics [130] (such as LiNO_3_:acetamide 22:78 eutectic, which was recently described as a valuable medium for electrochemical capacitors [131]) and the DES mixture already described (silver triflate:acetamide [79]), while Type III DES can form only BADES. The existence of these non-conventional melts was quite recently discussed regarding the preparation or modification of inorganic nanoparticles. These DES include chloride salts, which have been employed in three different preparations of titania-based nanomaterials: the nano-photocatalyst n-TiO_2_-P25@TDI@DES, which is highly recyclable and selective, was obtained by combining TiO_2_-P25 powder (70% anatase, 30% rutile) with a ZnCl_2_:urea 1:4 mixture using 2,4-toluene diisocyanate (TDI) as a bifunctional covalent linker. The nanocatalyst was employed successfully in the oxidation of benzyl alcohols to aldehydes and sulfides to sulfoxides [132,133]; the synergy between LADES and NPs was also exploited in the coupling between “Hierarchical TiO_2_” (H-TiO_2_) microspheres and FeCl_2_/CuCl_2_ urea in 1:4 mixtures to increase the reaction yields in the preparation of pyrrolidyn-2-one heterocycles [134] and by grafting the ZnCl_2_:urea 1:4 mixture onto magnetic ferrite nanoparticles to generate a DES@MNP homogeneous catalyst (which was found to be “magnetically recyclable” at the end of one-pot multicomponent syntheses [135]). Coming back to the synthesis of the nanoparticles themselves, some lanthanide-based type IV DES (Ln-DES) containing hydrated nitrates were prepared [136]. These mixtures show unusually low viscosity and surface tensions and the presence of fluxional oligomeric polyanions and polycations, and were employed as reaction media in the combustion synthesis of oxides; more recently, actinide-based type IV DES (An-DES) were obtained by mixing uranyl nitrate hexahydrate UO_2_(NO_3_)_2_·6H_2_O (UNH) with urea in different ratios (Figure 4), finding 0.2:0.8 to be the optimal UNH:urea mole fraction, with a quite low eutectic temperature of −5.2 °C. This liquid was employed to prepare UO_2_ nanoparticles through an optimized electrosynthesis path [137]. Some examples of reactions in LADES are given in Table 2.

Aside from metal nanoparticles, DES are finding wide application nowadays in the production of lignin and cellulose nanoparticles. A basic knowledge of this widely investigated topic, though not closely related to the focus of this article, can help in showing further examples of eutectic mixtures, particularly the newest natural biocompatible ones, that could somehow be adapted and employed for the synthesis of systems containing metals. Among the most recent examples worth mentioning are mixtures of choline chloride:glycerol (or ethylene glycol) with AlCl_3_ [138]; choline chloride with ethanolamine or polyvinyl alcohol [139,140]; and choline chloride coupled with carboxylic acids (such as oxalic or lactic acid [141]) or with inorganic salts (Gly-K_2_CO_3_) [142]), DES lactic acid-betaine [142], and menthol-based melts (e.g., menthol:dodecanoic acid [143]).

**Table 2 molecules-27-02045-t002:** Examples of reactions in type IV DES (LADES).

Solvent	Reagents/Path	Product	References
Lanthanide nitrate hydrate:urea 1:3.5	Mixing of cerium (III) nitrate hexahydrate, neodymium(III) nitrate hexahydrate, or praseodymium(III) nitrate with urea at various ratios; preferred ratio is 1:3.5	**Lanthanide oxides CeO_2_, Pr_6_O_11_, NdO_3_ + mixed carbonates**	[140]
UNH(UO_2_(NO_3_)_2_·6H_2_O):urea at various ratios	Mixing of UNH:urea at ratios 0.9:0.1, 0.8:0.2, 0.75:0.25, 0.6:0.4, 0.5:0.5, 0.33:0.67, 0.2:0.8, 0.1:0.9. Melting point of −5.2 °C for the 0.8:0.2 mixture	**UO_2_** NPs	[141]
ZnCl_2_:urea 1:4	DES: Urea (20.0 mmol, 1.200 g) and zinc chloride (5.0 mmol, 0.680 g). Covalent bonding of TiO_2_ to DES through 2,4-toluene diisocyanate (TDI). Dispersion of 0.5 g of TiO_2_@TDI NPs in DES with stirring at 100 °C for 18 h; washing with ethanol and drying at 60 °C under reduced pressure for 6 h	**Ti@DES** nanocatalyst	[136,137]

## 6. A Prototypical Synthesis

In order to show the potential and ease of obtaining inorganic nanoparticles in DES, we show a prototypical NP synthesis carried out in DES in the following paragraph. In particular, the co-precipitation of the mixed salt Fe_3_O_4_ (structurally FeO·Fe_2_O_3_) from Fe^2+^ and Fe^3+^ soluble salt solutions in Choline-Urea 1:2 DES (reline) upon the addition of sodium hydroxide will be shown. The procedure was first reported in [56] and was modified for this test preparation according to the following protocol:(a)Preparation of DES

Totals of 11.108 g of dry choline chloride (MM = 139.62, 80 mmoles) and 9.610 g of urea (MM = 60.06, 160 mmoles) were weighted separately and kept in closed containers. The two solids were then mixed in a sealable vial at room temperature. Quite rapidly, the two solids when put in contact formed a sluggish agglomerate that became more fluid and transparent with gentle heating (35–40 °C); see Figure 3, first panel.

(b)Preparation of the solution

Totals of 0.130 g of FeCl_3_ (0.8 mmoles) and 0.167 g of FeSO_4_·7 H_2_O (0.6 mmol) were weighed and added to 1.559 g of the prepared DES in a 25 mL beaker. The dissolution was accelerated by heating the mixture at 80 °C, resulting in an orange transparent liquid; see Figure 5, second panel. The total amount of water introduced into the system (by counting the hydration water molecules coming from FeSO_4_ and estimating those coming from the average absorbed water content of the two DES precursors (ChCl and urea) by weighing samples of the powders before and after a three day-long drying treatment at 50 °C in an oven) was well below the upper limit (10 moles of water per mole of DES), as identified in the neutron diffraction study by Hammond et al. [48], beyond which the microscopic features of pure reline were lost and the system turned into a three-component HBA:HBD:water mixture.

(c)Precipitation

A total of 0.187 g of NaOH (previously powdered in an agate mortar, 46.7 mmol) was added to the solution at 80 °C under magnetic stirring (600 rpm). After 20 min, a black precipitate formed; see Figure 5, third panel. The precipitate was washed with 20 mL of distilled water four times and centrifuged for 5 min at 3000 rpm after each rinse cycle. The excess of DES crystallized upon contact with cold water but was gradually solubilized. The final pH after the washing was around 7, signaling that the excess amounts of NaOH and urea had been washed out.

(d) Drying

The washed precipitate was collected from the vial, dried at 40 °C and finally rolled into a bead. The bead was put into an Eppendorf tube, and its ferromagnetism was qualitatively tested with a magnetic stir bar (Figure 5, fourth panel)

## 7. Conclusions

In this small contribution, an overview of the newest and most important features of Deep Eutectic Solvents in the field of nanoparticle synthesis is given. We demonstrate that these mostly benign and inexpensive reaction media can be employed in different types of wet synthesis, as well as in electrochemical preparations. The reported reaction schemes are often straightforward and do not require complicated set-ups or equipment. The inherent inhomogeneity of the liquid mixtures exerts a direct templating effect without the use of any additives, and nanosystems of different topologies/shapes can be obtained. This field is being actively developed, as seen by the ever-increasing number of articles published, and its huge potential will surely continue to be explored, in search of new materials with beneficial technological properties.

## Figures and Tables

**Figure 1 molecules-27-02045-f001:**
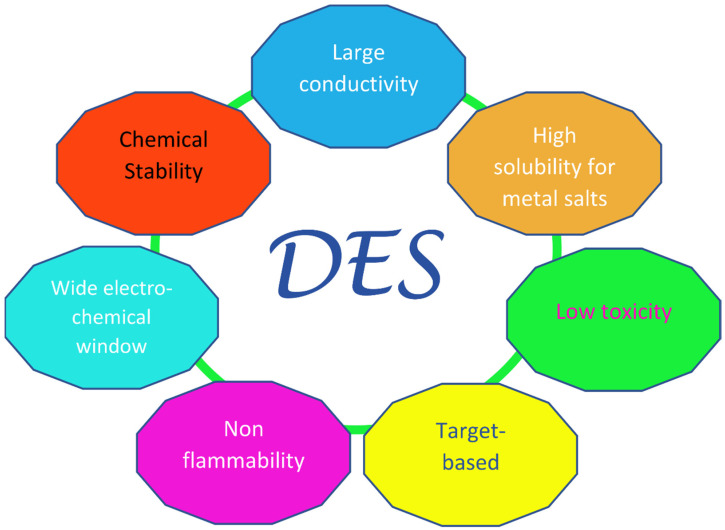
Pictorial diagram of the innovative properties of Deep Eutectic Solvents, inspired by Ref. [29].

**Figure 2 molecules-27-02045-f002:**
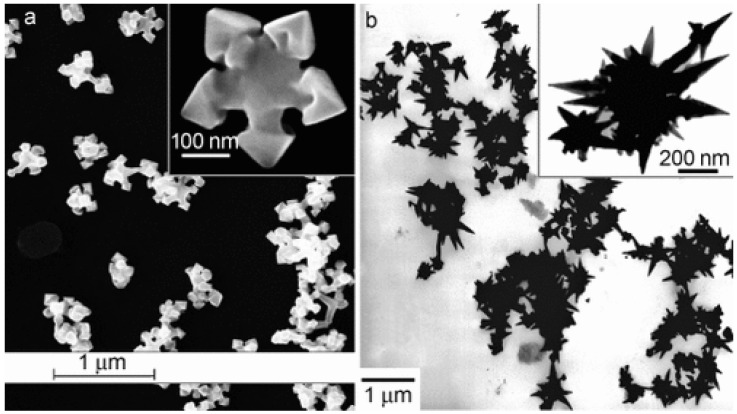
Deep Eutectic Solvents as green media in the synthesis of anisotropic Au nanoparticles (panel (**a**): flower shape; panel (**b**): nanothorn). Reproduced with permission from ref. [49]. Copyright 2008 John Wiley and Sons.

**Figure 3 molecules-27-02045-f003:**
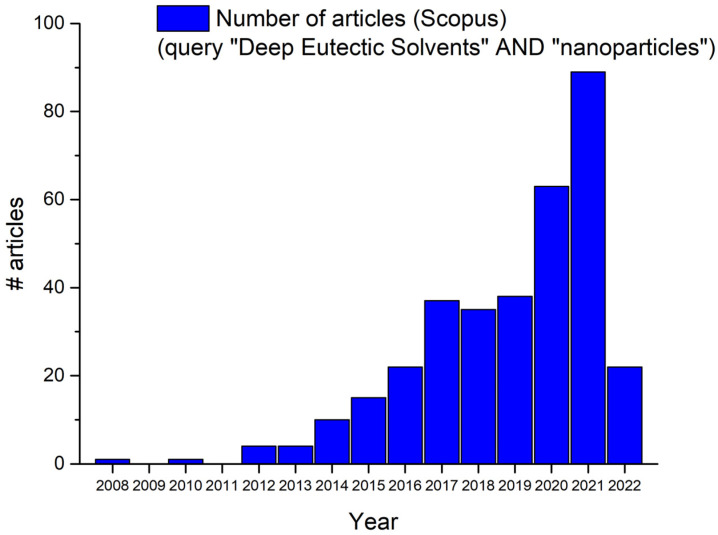
Number of published articles per year (2012–2022) corresponding to the Scopus query “Deep Eutectic Solvents AND nanoparticles” issued on 17 February 2022.

**Figure 4 molecules-27-02045-f004:**
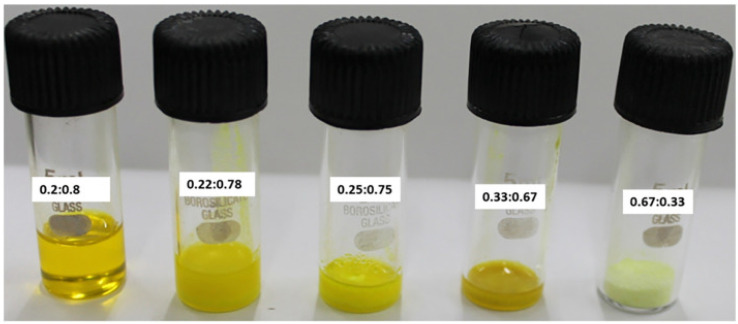
Examples of liquid mixtures of UO_2_(NO_3_)_2_·6H_2_O + urea DES, reproduced with permission from ref. [137]. Copyright 2021 Elsevier.

**Figure 5 molecules-27-02045-f005:**
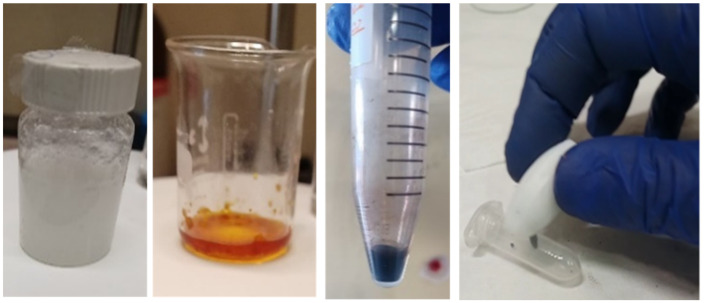
Demonstrative synthesis of magnetite nanoparticles. From left to right: Choline-Urea 1:2 DES forming at 35 °C; FeCl_3_ and FeSO_4_ salts dissolved in Choline-Urea 1:2 at 80 °C; Fe_3_O_4_ precipitate is formed on the addition of solid NaOH; after washing with water and drying at 40 °C, a magnetic solid bead is obtained.

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
