# Peer review of "Recent Advances in the Synthesis of Inorganic Materials Using Environmentally Friendly Media"

_molecules, 2022, doi:10.3390/molecules27072045_

Round 1

Reviewer 1 Report

Manuscript Number: molecules-1634783

Title: Recent advances in the synthesis of inorganic materials using environmentally-friendly media

In this review type of manuscript, the authors reviewed a summary of the numerous and most recent achievements in the area of synthesis and modification of inorganic materials, in particular, nanoparticles by focusing on DES media. The manuscript was written well and the authors did a comprehensive investigation on the defined topic. The order and style of context presentation were acceptable. There are just a few minor issues that I mentioned below. Therefore, By resolving the minor issues, I recommend publishing the manuscript in the Molecules.

·       In the caption of Figure 3, you should involve the exact date of search.  

·       In the section “Preparation of DES”, you must include drying for prepared DES and then measuring the amount of water content of that. 

Author Response

We warmly thank the reviewer for his/her positive comments and for the suggestions that are very helpful for improving our manuscript. In the revised version, all the text changes have been highlighted in yellow for your convenience. Regarding the raised issues:

·       In the caption of Figure 3, you should involve the exact date of search.  

We modified the caption of Figure 3 accordingly, introducing the actual date of the Scopus query (Feb 17th)

In the section “Preparation of DES”, you must include drying for prepared DES and then measuring the amount of water content of that. 

We thank the reviewer for this precious comment. We checked the water content of the reactants used and verified that the overall water content introduced into  the mixture was smaller than the water content "tolerated" by DES structure. as discussed in a neutron diffraction study (Hammond, O.S.; Bowron, D.T.; Edler, K.J. The Effect of Water upon Deep Eutectic Solvent Nanostructure: An Unusual Transition from Ionic Mixture to Aqueous Solution. Angewandte Chemie 2017, 129, 9914–9917, doi:10.1002/ange.201702486).

The article was cited and explicative sentences were added in paragraph 3 when discussing DES heterogeneity:

The preservation of the microscopical structure upon hydration was also demonstrated in a neutron diffraction study by Hammond et al. [48], who pointed out that the microscopic features of pure reline were maintained up to a 10:1 water to DES molar ratio (around 42 wt% H2O); beyond that limit, the system is better described as a three-component HBA:HBD:water mixture. 

Also the "preparation of the solution" paragraph in the last part was modified:

The total amount of water introduced in the system, by counting the hydration water molecules coming from FeSO4, and estimating those coming the average absorbed water content of the two DES precursors (ChCl and urea) by weighing samples of the powders before and after a three day-long drying treatment at 50°C in oven, is well below the upper limit (10 moles of water per mole of DES), identified in the neutron diffraction study by Hammond et al. [48], beyond which the microscopic features of pure reline are lost and the system turns into a three-component HBA:HBD:water mixture.

We hope that you find the modified version of the manuscript, containing these and the other changes made, acceptable for publication.

Reviewer 2 Report

            In this review manuscript, entitled “Recent advances in the synthesis of inorganic materials using environmentally-friendly media” authors have highlighted the role of DES in nanoparticle synthesis.  It is a significant review but, in some areas, the manuscript needs to be revised before it can be published. The specific suggestions are as follows:

  1. Specifically used templating agents for NP synthesis can be included in the introduction section (lines 65-68).
  2. The role of DES as reaction media, catalyst, or functionalization agent for NP synthesis can be highlighted briefly.
  3. An overview regarding advantages and challenges of using DES for NP synthesis can be included.
  4. “Figure. 35, first panel” line no. 407, can be corrected.

Author Response

We warmly thank the reviewer for his/her positive comments and for the suggestions that are very helpful for improving our manuscript. In the revised version, all the text changes have been highlighted in yellow for your convenience. Regarding the raised issues: 

Specifically used templating agents for NP synthesis can be included in the introduction section (lines 65-68).

More details were given in the sentences suggested:

Various types of systems were employed for this purpose, like, for instance, boric, citric and ascorbic acids [13], cholesteric liquid crystals [14] or different types of surfactants [15].

The preferred not to give details for the last study (ref.15), since it contains too many surfactants, and it was not easy to choose one example

The role of DES as reaction media, catalyst, or functionalization agent for NP synthesis can be highlighted briefly.

An overview regarding advantages and challenges of using DES for NP synthesis can be included.

A longer description of the advantages and challenges (drawbacks) of using DES was given at the end of paragraph 3 (DES in nanoparticle synthesis from the birth to 2020), including one more reference regarding the effect of viscosity on NPs growth:

Summarizing, the main advantages of DES that were readily highlighted shortly after their introduction as synthesis media and that were largely demonstrated during this initial time period, lie in their benignity towards ecosystems, in their highly tunable physiochemical properties and in their cheap preparation and handling. An additional very important and profitable feature resides in their intrinsic microheterogeneity, that is capable of conferring specific morphologies to the obtained nanoparticles. A possible drawback of the use of DES could be found in their moderate viscosity, that depends on the nature of its components, being lower for hydrophobic DES and larger for sugar-based NADES. Indeed, it has been shown that the decrease of mass-diffusivity caused by viscosity affects nanoparticle growth, and generally leads to NPs of larger size. [86]

“Figure. 35, first panel” line no. 407, can be corrected.

The error was corrected

We hope that you find the modified version of the manuscript, containing these and the other changes made, acceptable for publication.